# Student Participation: Issues for the Governance of Higher Education

Albertina Palma [1], Joaquim Mourato [2,*], João Vinagre [3], Fernando Almeida [1] and Ana Maria Pessoa [1]

[1] Center for Research in Education and Training, Polytechnic Institute of Setúbal, 2910-761 Setúbal, Portugal; albertina.palma@ips.pt (A.P.); fernando.almeida@ese.ips.pt (F.A.); ana.pessoa@ese.ips.pt (A.M.P.)

[2] Research Center for Endogenous Resource Valorization, Polytechnic Institute of Portalegre, 7300-555 Portalegre, Portugal

[3] Center for Innovation in Science and Technology, Polytechnic Institute of Setúbal, 2910-761 Setúbal, Portugal; joao.vinagre@estbarreiro.ips.pt

\* Correspondence: amourato@ipportalegre.pt

**Abstract:** The paper reports findings of a research project aimed at developing insight into student participation in the governance of higher education institutions. The project was carried out in two institutions in Portugal, analyzing numbers and forms of participation, identifying facilitators and constraints to participation, and analyzing the students' perceptions of their own participation. The study was carried out in the context of the European purpose of creating a cohesive European Higher Education Area (EHEA), and in light of theoretical perspectives of the sociology of public action. The concept of participation put forward by the Council of Europe in 2003 was mobilized in the sense that true participation involves making a difference in decision-making. The research objectives required both quantitative and qualitative data collection; therefore, a mixed-methods approach was adopted, including document analysis, interviews, and a questionnaire. The cross-analysis of the data collected enabled the researchers to characterize the participation of students in formal governing bodies and in other institutional contexts, identify facilitators and constraints to participation resulting either from the legal framework, the institutional culture, or personal contexts, and simultaneously capture individual perceptions of participation on the part of the students. The results enabled the authors to make a set of recommendations for political action both at national and institutional levels.

**Keywords:** education quality; educational policies; higher education; quality; student participation

## 1. Introduction

Student participation in the governance of higher education is a most relevant issue within the context of the creation of the European Higher Education Area (EHEA) piloted by the Bologna Process. Students have been involved in the Bologna Process from its very beginning, through their representative structure ESU (European Students Union), formerly ESIB (European Students International Bureau), by holding a seat as consultors in the Bologna Follow-Up Group [1]. The creation of the EHEA emerges in the context of globalization and a knowledge-based economy. In this context, the European Union intended to become the most competitive knowledge-based economy in the world [2]. Powerful worldwide pressures of economic competition were supported by neo-liberal policies and the new public management agenda, which eventually pervaded the Bologna Process [3]. This contributed to turn European universities from autonomous institutions into some kind of corporations or competitive business units [4,5]. As a result, higher education was opened up to society, with corporations and organizations being regarded as business partners with a say in institutional governance, that is, as stakeholders [6]. Stakeholders are "any group or individual who can affect or is affected by the achievement of the organization's objectives" [7] (p. 46). In this sense, students are considered stake-

holders, in the context of quality assurance in the EHEA, although they are a special kind of stakeholder because they are simultaneously internal and external to institutions [8,9].

Together with the European University Association (EUA), the European Association of Institutions in Higher Education (EURASHE) and the European Association for Quality Assurance in Higher Education (ENQA) students were appointed as part of the E4 Group, as well as other stakeholder organizations (Education International, Business Europe), and EQAR "to develop an agreed set of standards, procedures and guidelines on quality assurance, to explore ways of ensuring an adequate peer review system for quality assurance and/or accreditation agencies or bodies" [10] (p. 3). This working group outlined the European Standard Guidelines in 2005, which was revised in 2015 [11], and it is the guide to quality assurance within the EHEA to date.

The role of students within the ESG is defined as stakeholders: "stakeholders are understood to cover all actors within an institution, including students and staff, as well as external stakeholders such as employers and external partners of an institution" [11] (p. 5). Moreover, the ESG are aligned with the most recent trend in quality assurance, the so-called Third Generation of Quality [12], in that quality results from the consistent satisfaction and in the long run of the stakeholders' needs and expectations. No universal definition of quality is provided by the ESG. Instead, quality is viewed as resulting from the stakeholders' interaction in a given context:

> " ... stakeholders, who may prioritise different purposes, can view quality in higher education differently and quality assurance needs to take into account these different perspectives. Quality, whilst not easy to define, is mainly a result of the interaction between teachers, students and the institutional learning environment" [11] (p. 5)

Accordingly, students are regarded as central to the learning process, so the programs should be "delivered in a way that encourages students to take an active role in creating the learning process, and that the assessment of students reflects this approach" [11] (p. 8). Furthermore, the political discourse around the EHEA is that of valuing students' participation in the governance of higher education, on the assumption that " ... the reform of higher education systems and policies will continue to be firmly embedded in the European values of institutional autonomy, academic freedom and social equity and will require full participation of students and staff" [13], p. 1. Consequently, there is a commitment to "fully support staff and student participation in decision-making structures at European, national and institutional levels" [14], p. 2, and "continuing to involve students and staff in governance bodies at all levels" [15], p. 5. As for ENQA, concrete policies are to be supported, such as "ensure that existing legislation does not hinder students from participating in external quality assurance" [16], p. 2. On their part, students claim to be "or should be—full, or equal, members of the academic community" [17] (p. 6).

Despite the intentions stated across policy documents, students have claimed that participation is far from being a reality: "Both at national and institutional levels, it can generally be said that students aren't considered as full partners, neither in practise nor in theory [18] (p. 37). "The position of students has been empowered on the policy level, but the real involvement is not yet a reality" [19] (p. 7). In Portugal, the situation is not substantially different from that in Europe [20–22]. The Portuguese legal framework of higher education [23] establishes that students should sit on various national and institutional bodies, with institutions having the autonomy to create internal governing structures where students can be represented. This is, however, considered insufficient by, for example, the National Council of Education (CNE). The CNE, a mandatory consultative structure of the Government, made a recent important step forward with the publication of the Recommendation on the participation of young people in higher education [22]. The Recommendation highlights shortcomings in the legal framework while presenting some examples of good institutional practices aimed at promoting conditions for students to participate in the governance of institutions of higher education. Furthermore, in addition to the reduced number of students in the governance of higher education, as stated by law [23], a number

of constraints to participation have been identified in institutional contexts, such as: a lack of appropriate information, preparation, and support for participating in governance, along with better-equipped actors, a lack of appropriate and timely institutional information, perceptions of no worthiness and no impact of participation, limited time in the institution, and asymmetric pedagogical power relations [19,21].

Power is at the center of political decisions. Therefore, understanding factors influencing the participation of students in policy decision-making is paramount. In this study, we approach students' participation in Portuguese institutions of higher education from the perspective of the sociology of public action [24,25]. From this perspective, policy decision-making results from the action of multiple actors, ranging from international agencies to individuals, or carriers [26], who move across different social contexts and at different levels, either local, national, or supranational (idem). Such contexts, conceptualized as forums [27–29], involve several actors who dictate and carry different matrices and visions of the world and interact in power relations, from which hierarchies of actors emerge. Visions of the world include integrated systems of values, norms, algorithms, and images that shape policies [30–32]. Power can derive, among other things, from money, social position, prestige, influence, or the relative size of a group. In this sense, as important as ensuring that students are formally involved in local, national, and European decision-making structures, as in the case of the multilevel policies of EHEA, is examining whether and how the conditions that the students have enable them to really make a difference in those structures.

Making a difference is indeed the ultimate goal of participation. Therefore, in this study, the following concept of participation proposed by the Council of Europe in 2003 and followed by the European Youth Strategy [31] was adopted: "Participation and active citizenship is about having the right, the means, the space and the opportunity and where necessary the support to participate in and influence decisions and engage in actions and activities so as to contribute to building a better society" [33]. This is the same concept underlying the students' statement that: "Student Participation is not just a tool for students to complain about classes that they dislike, it is a fundamental way to shape learning paths and therefore to shape the society of the future" [34] (p. 14).

The researchers are, however, also aware of the complexity of participation and share Kahu's vision insofar as participation results from the interaction of personal, social, and political factors, and that there is a "need for projects that focus on narrower populations, including single institutions" [35] (p. 16). Thus, the study focused on two institutions of higher education with the broad aim of developing insight into levels and forms of student participation, by identifying facilitators and constraints, which come from the political framework, institutional culture, and personal contexts, and by analyzing the students' own perspectives on participation. Additionally, based on the findings, the study aimed to make recommendations for policy and institutional action at the national level.

The conclusions of the study point to somewhat disparate conceptions of participation on the part of the students. On the one hand, they barely participate in formal governing bodies and in elections of their representatives, although they say they value participation. On the other hand, some of them do not seem to complain about the situation. On the contrary, they say they enjoy taking part in projects and other extracurricular activities in the institution. Besides, they can solve their problems via direct interaction with professors and other staff. As for factors influencing participation, a number of factors were identified, both facilitators and constraints, which confirmed some of those identified by previous research, as mentioned above. Facilitators included positive feedback from peers, past experiences of participation, and the perceived value of participation by the institution. The main constraints were a lack of time, in addition to inadequate regulatory provisions, a lack of preparation, personal disinterest, agendas with no direct interest to students, and a lack of or insufficient information provided by the governing bodies about their activities. The conclusions are apparently in line with Kahu's conceptual framework [35] insofar as student participation is a multilevel issue involving personal, institutional, and

sociopolitical dimensions. Action is required within each level, and suggestions for tackling this was a third aim of this research project by making recommendations for Portuguese national and institutional policies.

## 2. Materials and Methods

The study adopted a cross-sectional research design, as defined by Bryman: "the collection of data on more than one case (usually quite a lot more than one) and at a single point in time in order to collect a body of quantitative or quantifiable data in connection with two or more variables (usually many more than two), which are then examined to detect patterns of association" [36] (p. 58).

The approach was considered appropriate for the research question that guided this study, as it aims to develop knowledge about characteristics, factors, and student perspectives in two Portuguese polytechnic higher education institutions (IP) (one on the coast and medium-sized, Instituto Politécnico de Setúbal (IPS), the other inland and small-sized, Instituto Politécnico de Portalegre (IPP)). As this involved collecting data on a rather large population of students, it was necessary to construct a representative sample of students, who answered a questionnaire. The questionnaire included multiple dimensions and indicators that were analyzed in order to find interrelated patterns.

Thus, the students' responses to the questionnaire made it possible to understand the value that students attribute to their institutional participation, as well as its characteristics and factors. The data thus obtained were complemented and reinforced by the data collected on the participation of students in electoral acts for governing bodies and in their participation in the respective meetings. This was achieved by collecting and analyzing data from institutional documents over the period 2015–2020. Official documents were also analyzed in order to establish the national legal context of the governance of the Portuguese HEIs as well as the internal regulations of the two institutions under analysis. All these data made it possible to build a comprehensive picture of the situation under analysis, which was based on representative sampling, in the case of the questionnaires, and on the scope of the information obtained on the electoral processes for the governing bodies and their functioning.

Nevertheless, it was decided to collect more in-depth information about the research object, applying semi-structured interviews to a small number of students. According to Bryman, in a cross-sectional research design, other techniques, in addition to the questionnaire, can be adopted, such as the semi-structured interview: "this will allow me to retain the conventional understanding of what a survey is while recognising that the cross-sectional research design has a wider relevance—that is, one that is not necessarily associated with the collection of data by questionnaire or by structured interview" [37] (p. 59). The semi-structured interview was paramount in the adopted methodological device, as it helped to interpret the quantitative data of the questionnaire, as well as to grasp the meanings that the students attributed to the levels and factors of their participation.

The data collection and analysis techniques [37–39] were framed by the theoretical perspectives and the context described in the Introduction Section, namely the power relations associated with the role of students as stakeholders within the EHEA, and the multilevel nature of participation. Techniques were used in temporal sequence, in three distinct phases. In each phase, the respective analysis led to results that provided content for the selection of information to be collected in the following phase. Finally, a global analysis of the results was conducted, by crossing the partial results. For that, a table was created containing categories resulting from the research objectives.

### 2.1. First Phase—Document Analysis

The research started by focusing on how student participation is addressed in the legal and internal regulatory documents, and secondly, on levels of participation in elections and meetings in internal governing structures. The aim was to establish the situation concerning the research topic, in the institutions under analysis.

Document analysis was of three types. Firstly, documents relating to the national legal and regulatory framework were examined, based on the Legal Regime of Higher Education Institutions (HEIs), along with the internal institutional regulations for the implementation of student participation, as provided for in the respective Statutes. An analysis of the content of the norms was carried out with a focus on the governing bodies where students have a seat, in light of the principles and guidelines on student participation outlined in policy documents.

Another type of analysis was based on quantitative data on student participation provided by the minutes of the electoral processes of the governing bodies and the academic unions, over the period 2015–2020. Minutes were requested from the presidencies of both institutions under analysis.

Finally, an analysis was carried out on the participation of students in the meetings of the bodies to which they were elected, in the same period (2015–2020) in the two HEIs, and on the topics included in the agendas of these meetings. Statistical analysis was carried out based on meeting agendas and meeting attendance lists.

This phase globally confirmed low levels of participation in governing bodies' elections and meetings and ended with three questions that were addressed in the interviews and in the questionnaires that followed:

- How interesting do students find the agendas of the meetings?
- From the students' perspective, how relevant is student intervention considered by the institution? Is it integrated in decisions?
- What influences students' decision to participate?

*2.2. Second Phase—The Interviews*

Interviews were used to obtain contextualized and in-depth information about the participation of students in each institution. It was deemed important to listen to students who, in recent years and/or currently, held positions in governing bodies considered by the legislation in force, as well as those who did not.

Sixteen (16) interviews were carried out with students from the two institutions and another one with the president of the National Federation of Students' Unions of Higher Polytechnic Education (FNAEESP), who studies at another institution. The aim was to distribute the number of interviews proportional to the number of students in each institution, so nine students from the IPS and five from the IPP were interviewed. The interviews included students who participated in one or more governing bodies and others who chose not to.

Regarding the selection of students with governing positions, one governing body was chosen in each IP and in each of its schools, to globally ensure its diversity. Then, the president of each of the selected bodies was asked to indicate three students. The first on that list was contacted by the research team and, in the absence of a positive response to the invitation, the following one was contacted. Regarding students without governing positions, a course was randomly selected from each of the schools, and in that course, a year and a list of students were chosen. The first of them was contacted via email, in which the study and the interview proposal were explained. Only in case of refusal were the remaining contacts made. Thus, 10 students with governing positions and 6 without positions were interviewed. Of those interviewed, 9 are female and 7 are male. As for students with governing positions, the majority are male (7 male and 3 female).

The semi-structured interviews were carried out based on a script organized into four blocks, with several interconnected questions, which also integrated the questions resulting from the document analysis, as listed above. The individual interviews were conducted by one of the team's researchers, with the assistance of another who, if necessary, could also intervene. All were carried out at a distance, using a digital platform, and were recorded with the interviewees' permission (with completion of a free and informed consent form). The duration of each interview varied between 30 and 60 min, and in exceptional cases, a little longer. For the treatment and analysis of the interviews, automatic transcription

software was used, with the text being heard and corrected by each of the researchers who were assigned the task. In the categorization of the responses, the previous grid of categories defined for the interview script was used. However, whenever necessary, the full interview was used to clarify doubts or add some important aspects.

### 2.3. Third Phase—The Questionnaire

The questionnaire incorporated the information resulting from the document analysis and from the responses obtained in the interviews. Based on this, the following dimensions were established, which were later included in the structure of the questionnaire:

- Valuing student participation.
- Institutional integration of student contributions.
- Participation facilitators.
- Constraints to participation.
- Modes of effective participation.
- Personal and academic description.

The questionnaire was constructed, made available, and answered through the IPS survey platform, based on the LimeSurvey application.

The anonymity of the answers was guaranteed, and the respondents had the possibility to declare their consent to participate in the survey. Those responsible for data protection at the IPS and the IPP as well as the ethics committee of the IPS were consulted in advance.

The population defined for the study were the students of IPS and IPP courses at degree and master levels, as well as students of higher education professional courses (CTESP). A stratified sample was constituted by school and course and by "clusters", by randomly choosing the Curricular Units (CU) of each year/course, with the respective students being considered in the selection of those who would respond to the survey.

The students' responses were collected in one of the CU classes chosen in the sampling process, from March to May 2022. The respective professor asked the students to answer the questions via mobile devices, showing a link and a QRcode that allowed access to the questionnaire.

A total of 919 responses were obtained, which was considered to ensure significant representativeness, since it was greater than a sample size defined with a confidence level of 99.7% and a sampling error of 5%, with a total of 825 students. Considering an increase of 20% to this last number of students to compensate for the lack of answers, response rates were obtained in the IPS and the IPP in the order of 90%. This allows considering the representativeness not only in relation to the two subpopulations, but also in relation to the population. Considering that there was an overrepresentation of responses in some schools, the representativeness per school had to be relativized.

The answers given by the students in the questionnaire survey were treated statistically, first in a descriptive and global way, and then looking for statistically significant relationships between the various responses and factors, such as school, age, gender, type of course, and course year.

### 2.4. Global Analysis of the Results

The analysis of the collected documents provided information that made it possible to outline the normative framework for student participation in the context of possibilities and limitations. In this context, it was possible to identify the levels of student participation in the electoral acts and meetings of the governing bodies of the institutions under analysis. Statistical analysis of the students' responses to the questionnaire provided an overview of the students' perspectives regarding the value they ascribe to their institutional participation, forms, and types of participation, and facilitating factors and constraints. In turn, content analysis of the answers to the interviews provided an in-depth view of the students' perspectives on participation.

Thus, after obtaining the partial results from each of the techniques, triangulation was conducted with a view to establishing data validation and complementarity. This was

achieved by creating three tables, each one corresponding to one of the research objectives: (1) characterization of the students' participation in governing bodies, (2) identification of facilitators and constraints (whether in norms or practices), and (3) analysis of students' perspectives on their participation in HEI governance. Each table was divided into three columns, with each column corresponding to each of the three techniques used.

Reflection and discussion on the interconnections of the contents inserted in each column made it possible to confirm their compatibility and complementarity for the interpretation of the data collected, with the aim of finding answers to the research questions.

## 3. Results

Cross-analysis of the data (see Supplementary Materials), as outlined in the previous section, confirmed the validity and complementarity of the results. These results are contained in three tables and described in the short texts below each one, under the headings corresponding to the objectives of the study.

### 3.1. Students' Participation in Governing Bodies

Students' participation, as illustrated in Table 1 below, can be characterized as follows.

**Table 1.** Characterization of the students' participation in governing bodies: summary of major results achieved by the different methods.

| Document Analysis | Interviews | Questionnaire |
|---|---|---|
| 1. As a rule, participation in electoral acts of formal structures is low. 2. Participation is lower (less than 10%) in elections that include all IPS students, which does not happen with IPP students (above 20%). 3. Higher participation rates in elections of smaller schools, in the IPS, as opposed to the IPP. 4. Distinct levels of participation in the institutional meetings (35–89% in the IPS; 30–75% in the IPP) | 1. Students with governing positions have higher participation in elections. 2. Reasons for abstentionism: (a) the existence, as a rule, of a single list for the bodies (presented very close to the electoral act); (b) the lack of communication between the students in the governing bodies and the others; (c) the ineffectiveness of institutional information; (d) a lack of democratic culture. 3. Parallel to formal participation, there is participation of an informal nature. 4. Types of participation not addressed in the study: institutional or research projects, curricular or other forms of interaction between curricular units and with the community. 5. Students generally feel welcomed, and that the institution is responsive to their presence and opinions. | 1. Low participation in formal structures. Still, informal participation with peers and with professors occurs. 2. Levels of participation in discussions with peers are higher. |

The data confirmed low voting rates in the electoral acts of governing bodies and student unions: less than 10% in the IPS and a little higher in the IPP. In the case of the IPS, participation was particularly low in the election for the General Council (the most important strategic governing body), probably because the topics of the meetings are of lesser interest to students.

In contrast to the low participation in voting, the few students who sit on governing bodies are very often present in the meetings or activities. This probably means that these students have an interested profile and are more committed to the functioning of the institution where they study. Moreover, students who hold governing positions are more involved than others in the electoral processes, namely voting, and consider themselves welcome in the bodies to which they belong. This may mean that institutions are likely to create favorable conditions to student participation.

The data collected also reveal that students have some possibility of expressing their opinions and having their contributions considered by the respective institution in informal contexts with colleagues, with or without professors and course coordinators. In interviews, the students confirmed this informal type of participation and revealed ignorance of other instances of governance: "I felt that the Director was . . . rather friendly and very concerned about their students" (JP). Therefore, in addition to formal participation in governing bodies, more invisible types of participation emerge, such as direct interaction with professors and course coordinators: " . . . I felt that the professors and the Director were responsive to me" (JP). This type of informal participation could explain why students say that their contribution is accepted, such as, for example, about pedagogical activity, social support, interaction with the community, and organization and functioning of services.

### 3.2. Facilitators and Constraints to Participation

Table 2, below, presents the facilitators and constraints to participation, as identified by the students. An interpretation of these is provided in the following text.

**Table 2.** Identification of facilitators and constraints (whether in norms or practices): summary of major results achieved by the different methods.

| | Document Analysis | Interviews | Questionnaire |
|---|---|---|---|
| **FACILITATORS** | 1. Higher student participation in Pedagogical Council (parity governing body) | 1. Students with governing positions are apparently well-informed about the governing bodies. 2. Experience of democratic participation outside the institution can be a facilitator. | 1. In general, all the factors listed in the questionnaire are valued by a very significant percentage of students. 2. Most valued facilitator outlined in the questionnaire: problem solving in direct interaction with teachers and staff (74.5%) 3. Interaction with the community is perceived as highly integrated. 4. Students acknowledge that some of their contributions are integrated by the institution (they are, however, perceived differently in the different schools and vary according to courses and students' profiles). |
| **CONSTRAINTS** | 1. Student participation in governing bodies was reduced by the latest Portuguese legislation [23]. 2. Addressing topics in meetings with no direct interest to students leads to student distancing. This was more significant in IPP (31% on average in IPP, 58% on average in IPS). | 1. There is a little dissatisfaction with the functioning of the bodies where students participate. 2. Students without governing positions barely know about governing bodies. 3. School and family education do not value participation; lack of time: work, transportation, schedules, etc. 4. Lack and flaws in the provision of institutional information; little value assigned to participation duties; little rotation of positions. 5. Also highlighted (50–60%) in the interviews is the lack of necessary knowledge for the task (and they feel that the Institution considers it, too). | 1. Most valued constraints outlined in the questionnaire: lack of time to perform the academic tasks (81.2%); lack of information and knowledge; institutional disinterest and devaluing of student contribution; students' disinterest; fear of retaliation. |

The current Legal Regime of Portuguese HEIs [23] reduced the number of students in the governing bodies of the polytechnic HEIs from 35% to 15%, which is not in accordance with the principles underlying the construction of the EHEA. This is especially serious regarding the presence of students in the General Council, the body where the president is elected and where action plans and budgets are approved. In this context, the minor

participation of students in HEIs' governance could lead to the perception that their opinion does not count and to a low participation in electoral acts for governing bodies, as this study showed.

Addressing topics in meetings with no direct interest to students leads to student distancing. This was more significant in the IPP (31% on average) than in the IPS (58% on average). Another important factor is the difficulty of accessing agendas prior to meetings, not only by the students who are members but also by the others, as this is a matter of their concern: "My opinion is that it would be a good thing to find some kind of device to give information to the students..." (JPC).

The most valued facilitators and constraints outlined in the questionnaire were: a lack of time to perform the academic tasks (81.2%), problem solving in direct interaction with teachers and staff (74.5%), a lack of information and knowledge, institutional disinterest and devaluing of student contributions, students' disinterest, and a fear of retaliation. As for the interviews, the main constraints were the academic workload: "it isn't possible, in my opinion, to be a 100% association leader and a 100% student. I don't think so, it isn´t possible" (JPP), and pressure from parents: "My dad and my mom said, before going to school, remember you are a student, you need to have good marks and then go to work" (AF). Another constraint was the long commute time from home to school. On the other hand, students felt they could solve their problems informally with professors and services, with no need to resort to more formal participation: "I think that if I had a problem, I would talk to my course director. I think he would be the first person to talk to" (BB). Lack and quality of information about the institution were also referred to. Students referred to the inadequacy of the formats for dissemination of information as demotivating and limiting factors for greater participation in electoral acts (expressed more affirmatively by students without governing positions).

Additionally highlighted (50–60%) in the interviews was the lack of necessary knowledge for the task (and they felt that the Institution considers it, too). Students who do not have governing positions say that these positions are not for them: "the way he speaks is so political for me, I can´t do it".

From the interviews carried out, the idea arose that there is a big difference between students who occupy institutional positions and those who do not, in terms of information and civic awareness. This difference between profiles creates reproduction mechanisms in the occupation of positions, in which those who assume them tend to also assume others, generating the idea of small groups of students involved and that the functions are "rotating" between them. This logic has the effect of a tendency for the occupation of positions to be successively determined by students who already know each other and who in a way have established friendly relations, and it creates the idea that they are not accessible to everyone and that the choice ends up being made inside a restricted core. However, there are cases that demonstrate that a student managed to make a list and be elected without this "succession" effect.

### 3.3. Students´ Perspectives on Their Participation

An interpretation of the students' perspectives on their participation is provided after Table 3, below.

Regarding integration of students' contributions, this was perceived as lower by master and evening degree course students (they feel less heard, and their opinions less considered). Older students tended to overvalue constraints to participation, probably due to their living conditions and difficulties in articulating academic requirements with those of the job and personal and family life, as well as their smaller and more irregular presence at the institution.

In contrast to the reduced levels of participation in electoral acts and a legal framework that minimizes their presence in the governing bodies of HEIs, the perspective of students on their participation in governance of HEIs was almost unanimously recognized as very important. This means that these students consider it very important that their

contributions be considered in relation to a set of topics directly related to the quality of their academic path, in accordance with a general principle of the HEIs: the need for students to be heard and their opinions and proposals to be considered.

**Table 3.** Analysis of students' perspectives on their participation in HEI governance: summary of major results achieved by the different methods.

| Interviews | Questionnaire |
| --- | --- |
| 1. Students who hold positions are, in general, satisfied with the functions they performed and the way in which they did so.<br>2. Suggestions related to information are not entirely consistent. Some students consider that information should become more digital and closer to youth forums, abandoning traditional formats based on paper and institutional email. However, others seem to value closer institutional communication.<br>3. Students consider that the way institutions conceive and disseminate information is inadequate.<br>4. The forms of organization and work within the bodies can still be improved, although they are not the object of strong criticism.<br>5. Some students value participation in institutional projects (Eco Schools, for example), in research projects, in initiatives, such as, for example, open weeks and/or course dynamics and/or curricular units (or even individual ones) in the community (solidarity campaigns, support for community institutions).<br>6. Direct interaction influences the students' perspectives on the need to take part in institutional bodies and on their role in them. | 1. Students value participation: on average, 90% consider it as important or very important.<br>2. Low participation in formal structures, due to the importance given to personal contacts in solving academic problems. |

This table does not include document analysis because this method did not provide any information on the topic

The students interviewed who held governing positions confirmed the appreciation of their institutional participation and/or in the academic union, not only for their contributions and in the representation of students' interests, allowing the "voice of students" to be heard in the institutions, but also for allowing them to acquire new knowledge and development of social and personal skills. They considered that these soft skills will be very helpful to them in the future in various areas of their lives, both personally and professionally. They emphasized the opportunity of sharing and discussing experiences and perspectives with colleagues and professors. These students were generally satisfied with the functions they performed, and with the way they did it, considering that they had an impact on the body they were part of.

## 4. Discussion

After collecting and analyzing the data obtained from each of the three techniques and their triangulation, two interconnected issues emerged, which will be discussed below, and will serve as the basis for the proposals and recommendations of the study.

### 4.1. Low Participation in Institutional Organizational Processes and Governing Bodies

The current Portuguese law [23] significantly reduces the representation of students in the governance of HEIs, compared to previous regulations. This signals an effective devaluation of the participation of students, making them a minority group in the most important institutional structures. Thus, the students' capacity to propose agendas more in line with their own interests, to assert their points of view, as well as their impact on decisions [40], are compromised. The research data showed that, as a rule, most of the topics under discussion are not of direct interest to students, not motivating them to participate, as the data also showed.

A conclusion can be drawn that regulations established by law are not consistent with the principle that students are considered partners in governance as they do not have the same conditions to participate as other stakeholders. Furthermore, due to flaws of

institutional information, which were also identified as constraints to participation, it is possible to conclude that the student representatives do not have the possibility to outline the agendas, nor does the average student have timely access to meetings' agendas or minutes. If, in addition to this, we consider the lack of knowledge and of appropriate preparation that students have to successfully carry out this type of task, as was also identified, we can conclude that the ability to participate in these important forums is significantly lower as compared to the power and influence that others hold in the hierarchy of actors [30].

This research has an institutional focus that deserves to be highlighted. Evidence provided made it clear that institutional culture can constitute an obstacle to participation. Information and communication flaws emerged as the most striking features of this. Failures that were mentioned included the content and availability of information about the governing structures, namely their mission, functioning, and activities, as well as all the electoral processes and procedures, from which students distanced themselves. Students also kept themselves away from the daily activity of governing bodies and did not contribute to their decisions. Other factors emerged that constrain participation, such as the lack of incentives. When incentives are present, especially from peers, students stated that they would be more easily motivated to participate. Likewise, the lack of interest and devaluation of participation by the institution seemed to play an important role.

It does not come as a surprise that the classroom and the class context appeared as the center of student activity, which emerged as satisfactory as far as sociability and needs for support are concerned but is disconnected from the academic community.

The data thus enable us to conclude that the HEI environment does not ensure the necessary conditions to develop an institutional project in a partnership where the students' perspectives are integrated so that students can make a difference in decision-making. On the contrary, the environment described above reinforces the personal constraints that the students refer to and constitutes a powerful constraining factor to participation.

### 4.2. The Different Forms and Perceptions of Participation

In contrast to the weakness of student participation in the functioning of bodies and their electoral processes, either due to the under-representation of students in their constitution, or due to the manifest lack of interest in electoral processes and in monitoring their activities, it appeared that participation is valued by students and is even an institutional reality. This may include institutional or research projects, curricular or other forms of interaction between curricular units and the community, and above all, direct contact to solve individual problems. These varied forms and contexts of participation, in addition to governing bodies and academic unions, seem to be disseminated in the relationships established between students, professors, and course coordinators.

This type of intervention that the students mentioned when asked about reasons for the weak participation in governing bodies, and that some of them consider they do not need, somehow resonates in the study carried out within the scope of the STUPS project [20]. As mentioned above, in this study, the classroom emerged as an intervention space, although with less weight than the intervention within the scope of the Academic Union. This understanding of participation can be addressed in relation to other data from this project, which are associated with constraints. These are mainly of a personal nature, but some are institutional, and specifically pedagogical, which were mentioned both in interviews and in the questionnaire. In addition to personal disinterest in participating in governing bodies, obstacles that emerged included the time needed to complete academic work, commuting time from home to school, family pressure to finish the course, and professional work that leaves no time for anything else.

On the other hand, many students did not see low participation as a problem. On the contrary, they think that individual contact with professors and/or other staff is the best way to solve their academic problems. Participation is thus viewed as a way of resolving personal issues. Within a framework of quality assurance, it is, however, relevant

to establish how this is an effective way of meeting the needs and interests of this group of stakeholders. It might work in cases where professors and services are responsive, but obviously some might not be. It is relevant, in this regard, to introduce another constraint to participation pointed out by previous research [19,21] and confirmed by this study, which is the fear of retaliation. The fear of retaliation, present in the students' discourse, can interfere with their will or need to raise questions, due to the fear of possible negative impacts on their assessment by professors and on the completion of their courses. This form of participation is based on close relationships between students, professors, and all staff, which is something positive. However, it does not in itself have an impact on decision-making, as highlighted by the students' representatives at the European level when they stated that participation is not just a tool for complaining, it is rather a fundamental way of shaping learning and future society [34] (p. 14). In this sense, it becomes important to address the question of whether and how it will be possible to move from the classroom level, where students feel more willing to discuss their learning path, to an institutional level where decisions are made. Thus, the data confirmed the complexity of participation and the difficulty of disentangling the factors that influence it. Student participation is a function of a composite and interactive context constituted by the political framework, the institutional culture, and the students' personal context.

*4.3. Recommendations*

The study's conclusions, based on the multidimensionality of the identified factors, allowed formulating a set of recommendations aimed at ensuring the best conditions for student participation. The best conditions depend on changes that must be introduced at national and institutional levels, capable of motivating and empowering students to participate.

Firstly, it is crucial to introduce changes to the RJIES [23] in order to eliminate the inconsistency between the valuation of student participation claimed in European policy documents and the reducing student representation in the governing bodies of the Portuguese HEIs. Most importantly, institutions must incorporate student participation as a strategy for action.

In this sense, institutional meetings should be organized around previously announced agendas so students can better prepare for discussions and propose topics for discussion themselves. Moreover, all institutional information must be available, including not only the functioning and activity of the organization, but also the agendas and resolutions of the governing bodies, by making use of new digital tools and networks. Furthermore, considering the informal types of participation that the students mentioned, it will certainly be possible, in collaboration with students' unions, to find ways of interconnecting it with the formal activity of governing bodies in a bottom-up flow of communication. These dynamics can reduce the distance between the level of the most used form of participation and the level of decision-making, thus introducing new ways of contributing to the institutional quality assurance system. Finally, and because HEIs are educational institutions and quality assurance systems reconfigure students as protagonists of their learning paths, training for participation and citizenship must constitute a priority institutional mission, by promoting knowledge about the democratic functioning of the organization and the circulation of institutional information.

In short, it is imperative to promote skills suitable for participation, such as critical and self-critical sense, the sense of the plural collective and the ability to work in groups, analysis and problem solving, and robust argumentative skills. Participation as an unquestionable democratic value must be present in the institutional environment and constitute a primordial pedagogical mission, aiming at building a future that integrates the perspectives of today's students, who will be professionals and citizens tomorrow.

## 5. Conclusions

This study stemmed from the recognition of the importance of student participation in HEIs within the framework of the construction of the EHEA. In this context of higher education reorganization, students are regarded as key-actors in governance and stakeholders within quality assurance processes.

In parallel, as protagonists of HEI governance at different levels, the experience and action of the research team members are based on valuing student participation and effectively listening to their opinions as an essential contribution to internal quality assurance. However, in their daily practice of performing their different duties, they have witnessed difficulties in mobilizing students to participate in governing bodies and in academic unions, whether in organizing lists of candidates or participating in meetings and other related activities.

The questioning of this reality felt in institutions' daily life was an essential starting point for the present study. This started with research about student participation in other contexts and on the relationship of this with the European political and educational framework of which Portugal is a part. Despite that this framework is based on an integrative principle and promotes participation as an essential value for the construction of the EHEA [41], evidence exists that in practice, this does not happen satisfactorily [19]. For the purpose of this study, participation means the capacity of influencing decision-making to build a better society [33].

These questions led to the definition of the three axes of analysis: characterization of student participation in the governing bodies where they have a seat and in the student unions over the past five years, in-depth understanding of facilitators and constraints to participation, and analysis of students' perspectives on their own participation.

The conclusions, among other things, point to a diversity of forms and types of participation that are used interchangeably by students, ranging from individual informal interactions with professors and services for solving individual academic problems, to the status of a member and a representative of students in the governing bodies and structures where policy decision-making takes place. Depending on the conception that one has of participation, the need of and how to participate is differently felt. Nevertheless, following the determinations of the Council of Europe, a democratic society such as the European Union in which we live is based on principles, assumptions, and rules that everyone should be conscious of and where everyone should have conditions to intervene for the good of all. How are educational systems and institutions fulfilling their role of preparing future citizens for this? This is an issue for education, particularly for higher education, which emerges as a challenging topic for future research.

This is an issue for education, particularly for higher education, which emerges as a challenging topic for future research. This is only a limited contribution to the comprehension of student participation as a multilevel issue. Firstly, the unit of analysis is small within the Portuguese higher education system, and secondly, the inquiry did not include all the institutional actors who have a say in the matter. In future research, staff, particularly professors and institutional top governance leaders, can be involved in order to integrate their perceptions. The study can, however, be enlarged in the national context and replicated in other contexts, particularly in other European countries, as they share the political educational framework of the EHEA. Furthermore, a comparative international study would add important value to an in-depth development of knowledge on the topic.

**Supplementary Materials:** The following supporting information can be downloaded at: https://www.mdpi.com/article/10.3390/soc13050115/s1.

**Author Contributions:** Conceptualization, A.P., J.M., and J.V.; methodology. F.A.; investigation, J.M., F.A., A.P., and A.M.P.; writing—original draft preparation, A.P., J.V., and J.M.; writing—review and editing, J.V., J.M., and A.P.; project administration, A.P.; funding acquisition, J.M. All authors have read and agreed to the published version of the manuscript.

**Funding:** This research received no external funding.

**Institutional Review Board Statement:** The study was conducted in accordance with the Declaration of Helsinki and approved by the Ethics Committee of the Instituto Politécnico de Setúbal (CE-IPS No. 07/2022).

**Informed Consent Statement:** Informed consent was obtained from all subjects involved in the study.

**Data Availability Statement:** https://ipsetubal-my.sharepoint.com/:f:/g/personal/joao_vinagre_e stbarreiro_ips_pt/ErqjU7uDU8hAuxyZddzbrFAByJP3tgzhSWcnqm_bJs8nIQ?e=x0KbIY (accessed on 28 March 2023).

**Acknowledgments:** The authors acknowledge the work carried out by the research team of the project "Student participation and quality of HEIs (PEQUES)" for providing the basis for this article: Bruno Fragueiro, Carla Cibele Figueiredo, Filipe Fialho, Inês Silva, João Pedro Pereira, Maria do Rosário Rodrigues, and Rodrigo Teixeira Lourenço.

**Conflicts of Interest:** The authors declare no conflict of interest. The funders had no role in the design of the study; in the collection, analyses, or interpretation of data; in the writing of the manuscript, or in the decision to publish the results.

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
