# Peer review of "Student Participation: Issues for the Governance of Higher Education"

_societies, doi:10.3390/soc13050115_

Round 1
Reviewer 1 Report
I really welcome this article. It clearly explains the reasons why student participation in governance structures is an important topic to examine. I think it also raises some interesting points that need addressing in future policy and practice. Student participation is seen as desirable, but to what extent is it really supported by institutions.
Important observations are made by this article for example:
Lines 110-112 “Student Participation is not just a tool for students to complain about classes that they dislike, it is a fundamental way to shape learning paths and therefore to shape the society of the future”. [32]
Lines 365-366 it is possible to conclude that neither the students´ representatives have the possibility to outline the agendas nor does the average student have timely access to meetings’ agendas or minutes. Good point students need to able to address issues they think are important otherwise the power is always with the institution.
Line 389 It is really interesting that they want to solve local problems but not instigate structural change. Could this be a need to teach active citizenship in the curriculum?
I have a couple of points where I would like further clarification as someone who is not familiar with the Portuguese educational context.
Line 65-66 Bernstein’s (2000) model of democratic education seems relevant here, maybe something to look at?
Line 101 Social and cultural capitals may enable some social groups to participate or others to be disadvantaged. The reasons for non-participation seem very complex and multi-layered, maybe point that out early on.
Line 139 ‘ Finally, a global analysis of the results was carried out, by crossing the partial results. I was not sure about the meaning of this sentence does it mean triangulation, or the cross-referencing of partial results?
Line 194 Could you clarify what the theoretical perspectives were ?
Line 284, Were there any limitations to the study, no research method is perfect?
Line 345 The issues with Portuguese legislation be explained in more detail for an international readership?
The article is generally well written and structured. The tables perhaps could be formatted a bit clearer. I think this is a valuable article and the authors may want to reflect on what future research needs to be done on student participation, maybe an international comparison?
Author Response
The authors welcome your comments, which are regarded as an opportunity to improve the article. We have tried to respond as best as we can.
We don´t disagree with the idea of introducing Bernstein model and will consider it in a next publication.
We agree about the multilevel nature of participation, and this is included in the project results. Thus, we have introduced the issue in the conceptual framework. On the other hand, the issue of disadvantaged groups is not addressed by the project.
In chapter 2 we have tried to clarify the process of triangulation as a method to validate and complement data and results.
We have clarified this point in the new version of the article.
We consider it difficult to go into much detail, but included some new information that may help.
The tables were improved, as suggested. We added a paragraph with suggestions for future research.
Reviewer 2 Report
This paper presents an interesting topic. It mentioned that higher education students’ participation on in the governance of higher education institutions. We all understand that students who born on the Internet era are seen as a source of power to improve the performance of social affairs. This study presents the steps of the strategic development in a sustainable support of documentation analysis and adopted a rigorous method of using quantitative and qualitative in data analysis. These conclusions should help us revisit our priorities in terms of the relative efforts in student participation and to make a set of recommendations for political action both at national and institutional levels.
Overall, the method is convincing and this paper can be accepted.
Author Response
The authors appreciate your review. Nevertheless, some improvements have been introduced as a result of in-depht analysis.
Reviewer 3 Report
Dear Authors,
After reviewing the article, I will proceed to list some comments so that you can improve its quality since I consider that the article is not subject to publication.
First, on the theoretical framework:
- I believe it is necessary to go a little deeper into the justification of the relevance of the study.
- It is not enough to present the theory, it is necessary to relate it, to create composite, processed and coherent content.
Second, on the materials and methods:
- The method of the study is not specified. To say that it is cross-sectional does not imply reporting the methodology. If it is a mixed design, what type of design is it? What phases does it present? There is no appropriate methodological description. What is the explicit general research question that frames the study being a mixed method? What are its parts? What objectives or questions are you going to answer in each phase? Employing two data collection techniques does not make a study mixed.
- What type of analysis has been performed on the documents? Do you have a structure of issues and thematic statements that structures your content analysis (early data reduction)?
- - A cross-analysis of the data is presented but no clear structure is provided to organize the data from the beginning. The term cross-analysis is used when I believe that a triangulation of data has been performed.
In third place, in relation to results:
- Along the same lines, the tables presented in the article are quite chaotic. It would be advisable to try to offer this information in a more convenient, aesthetic and clear way.
- I understand the density of information, but would it be possible to present them in a different way? An early reduction structure of the data, the use of visual representations of the subject matter is feasible to improve its presentation and final understanding.
- Try to eliminate the enumerations. They are empty of content. It is necessary to provide explicit, coherent and integrated reflections of the information.
Fourth, regarding the discussion and conclusions
- I strongly recommend add the limitations of the study.
- I recommend revising the conclusions, it would be positive to include more integrated information and not so enumerated.
I hope you will be able to attend to these considerations, which are always aimed at improvement.
With my best wishes.
Author Response
The authors welcome your thorough comments, which we regard as an opportunity to improve the article. We have tried to respond as best as we can.
The theoretical framework was revised in order to reinforce the rational of the research process.
We have attempted to clarify and better substantiate the methodology adopted. The research questions and objectives were highlighted and an attempt was made to explain the interconnection of the phases of the inquiry and the techniques used.
The tables have been changed.
Enumerations have been eliminated and limitations to the study included..
In addition, we have tried to improve the English language.
Round 2
Reviewer 3 Report
Dear Authors,
Thank you for your response.
I consider that the changes made have been sufficient to reconsider the publication of the article. It is true that it is necessary to revise it again before the final publication.
It is necessary to revise it:
- The tables, if they provide information on the phases, need to be ordered, as in table 1, where the questionnaire column is placed before the interview column.
- There are formatting errors, as in item 4.4, which does not appear in the corresponding place. I also recommend not providing the recommendations in a numbered form.
- Review the language throughout the document and take care of the expression, there is an excess of long sentences.
With my best wishes to the whole team,
Author Response
Dear Reviewer
Thank you for your helpful comments that once again contributed to improve the article.
The language was revised trying to eliminate long sentences.
The enumerations in the recommendations were eliminated and the tables were changed according to comments.